# Membrane Removal of Emerging Contaminants from Water: Which Kind of Membranes Should We Use?

**DOI:** 10.3390/membranes10110305

**Published:** 2020-10-25

**Authors:** Magda Kárászová, Mahdi Bourassi, Jana Gaálová

**Affiliations:** 1Institute of Chemical Process Fundamentals of the CAS, v.v.i., Rozvojova 135, 165 00 Prague, Czech Republic; karaszova@icpf.cas.cz (M.K.); bourassi@icpf.cas.cz (M.B.); 2Faculty of Science, Charles University, Institute for Environmental Studies, Benátská 2, 128 01 Prague 2, Czech Republic

**Keywords:** dense membranes, emerging contaminants, water quality

## Abstract

Membrane technologies are nowadays widely used; especially various types of filtration or reverse osmosis in households, desalination plants, pharmaceutical applications etc. Facing water pollution, they are also applied to eliminate emerging contaminants from water. Incomplete knowledge directs the composition of membranes towards more and more dense materials known for their higher selectivity compared to porous constituents. This paper evaluates advantages and disadvantages of well-known membrane materials that separate on the basis of particle size, usually exposed to a large amount of water, versus dense hydrophobic membranes with target transport of emerging contaminants through a selective barrier. In addition, the authors present several membrane processes employing the second type of membrane.

## 1. Introduction

Emerging contaminants (ECs) are once again raising the question of water quality [1,2,3,4,5,6,7,8] due to the serious effects of long-term contact, sometimes even at very low concentrations in drinking or surface water. Such amounts have, for many years, been either neglected, difficult to measure or unmeasurable by available analytical methods [9,10,11]. Therefore, there is naturally a time lag between the identification of a particular contaminant and its appearance in any form of directive regulating water quality. The European directive from 1998 related to drinking water quality, 98/83/ES, covers no endocrine-disrupting compounds, pharmaceuticals or personal-care products, but the new directive from 2017, 2017/0332 (COD) includes perfluorinated compounds and three endocrine-disrupting compounds (beta estradiol, nonylphenol and bisphenol A). The directives for surface water 2008/105/ES, dealing with environmental quality standards for 45 chemical substances, contain 14 endocrine-disrupting compounds according to the European database of endocrine-disrupting compounds (commission document SEC (2007) 1635), still does not include pharmaceuticals or personal-care products. Another approach to the correction of the situation is presented in the watch-list of monitored chemical substances in the European Commission implementing decision 2018/840 (repealing Commission implementing decision (EU) 2015/495). The list contains 18 contaminants of emerging concern: hormones, macrolide antibiotics and neonicotinoid insecticides included. The substances on the list are subject to monitoring in European countries and could potentially be included in their water directives. Although the process of assessing the contaminants and their inclusion in the legislation is a rather complicated and slow one, it is already obvious that technology for removing these compounds from water will soon be absolutely necessary. Emerging contaminants (ECs), however, refer to a group of substances with very different properties, from their molecular weights to their saturated vapor pressures. To illustrate the diversity of the group, Table 1 shows several physical and chemical properties of selected ECs.

Notwithstanding the fact of dealing with sometimes hard to measure soluble amounts in water, Table 1 clearly illustrates the main issue of removal of ECs from water, their diversity. That is why no universal method for their removal from water exists. Even narrowing the area of interest to, for example, pharmaceuticals, does not solve the problem because, for example, ibuprofen has dramatically different water solubility than paracetamol but they are both in the same group, called NSAIDs (nonsteroidal anti-inflammatory drugs). Different interesting reviews dealing with methods for removal of ECs [12,13,14,15,16,17,18,19] have common the conclusions: “The method was found to be quite effective, but…” The only thing common to ECs is the fact that they consist of carbon, hydrogen and oxygen. Yet, because wastewater treatment plants are not able to completely remove ECs [20,21,22,23,24], a strong effort is being made to find the right technologies.

Yang et al. [15] presented the following prospective methods for removal of ECs from wastewaters:Membrane filtration; specially NF and RO due to very small pores (<2 nm).Granular and powdered activated carbon adsorption for rapid filters and seasonal uses.Advanced oxidation processes recommended as extra treatments especially for drinking water quality improvement.

Wang et al. [13] mentioned also:
Graphene and graphene oxide adsorption due to their high and functionalized surface areas.Adsorption on carbon nanotubes with high adsorption performance and physicochemical properties (despite high costs).Biological degradation, the most used technique, requires space and time but is energy efficient.

Some alternative methods may be added:Constructed wetlands gathering a combination of natural processes: sedimentation, microbial hydrolysis and photolysis degradation [25,26,27].Metal organic framework adsorption (relatively new technology) [28].

The number of sorbents used is still increasing [14,29,30,31,32,33,34]. Especially carbon-based materials [31,33,34] are mentioned, but we can find also carbon nanotubes [12] and metal organic frameworks (MOFs) [28,35]. The hottest trend seems to be the use of plant-based coagulants [36], especially proteins of *Moringa olifeira* seeds [37]. New methods are also the subject of several recently accepted patents [38,39,40,41,42,43].

All of these methods have advantages and disadvantages. For example, advanced oxidation processes (AOPs), which are generally considered to be the most effective and universal, employ oxygen or free radicals to initiate the oxidation of undesired compounds [44,45]. The reaction should yield a less dangerous product or CO_2_ and H_2_O (total mineralization). AOPs and advanced catalytic oxidations are a very complex group of processes including ozonation, Fenton and foto-Fenton reaction, sonolysis, catalytic wet air oxidation processes, combined AOPs etc. [17,46,47,48,49,50,51,52]. AOPs show good ability to remove different types of emerging contaminants from water, but the problem is the possible toxicity of the byproducts of the reactions, which may be more toxic or biologically active than the original compounds. There may be a great difference between the results obtained with synthetic wastewater samples and real wastewater samples because real wastewater is a complex system and identification of the oxidation by-products may be difficult. The result of the application of AOPs for removal of ECs is influenced by the pH of the solution, the matrix, the presence of natural organic matter in the matrix and the dose irradiation. Carbon nanotubes (CNTs) [12] have a high adsorption potential and are better defined than activated carbon, but their adsorption is influenced by the chemistry of the solution, the surface chemistry of the CNTs, the type of packing etc. Besides that, there is a problem of the high price and possible toxicity of CNTs; to substitute one toxic component in the wastewater with another one is undesirable.

Membrane technology suitable for ECs is problematic because of the diverse physicochemical properties of materials and the wide range of parameters to combine for separation enhancement. Different kinds of membranes, namely polymeric, hybrid, inorganic or supported, implement almost any material or chemical that may be required for testing. For example, coating allows modification of the membrane surface, applying a molecular layer in order to change surface characteristics of the membrane [53]. Appropriate separation mechanisms and target pollutants narrow the choices of the membrane [54]. For example, a steric exclusion separation membrane should have pores smaller than the target pollutant. Moreover, modifying the membrane surface with a target compound may improve the separation efficiency. Experiment conditions such as pH, composition, temperature, water movements or pressure of the feed play major roles in separation. Usually, all mechanisms contribute to separation at once [55]. Notwithstanding, how intense each contribution may be, the separation mechanism may change during the process due to target pollutant effects. For example, an electrostatic repulsion membrane, based on pollutant rejection, may lose its surface properties during separation because of pollutant adsorption on the surface. This would lead to pollutant diffusion and cause the membrane to end up in permeable state [53]. The phenomenon is well known as the breakthrough curve of the membrane. Not only membrane separation tries to achieve 100% pollutant-free water, but full scale process application, energy evaluation (for membrane synthesis, and separation processes), the membrane’s material cost and environmental assessment, must be taken into consideration. For illustration, reverse osmosis (RO) showed higher performance in rejection of ECs, reaching 100% in optimal conditions, compared to nanofiltration (NF). However, environmental assessment of these two filtration technologies showed that NF is much more ecofriendly compared to RO because of reduced energy demand [18].

The presented paper focuses on membrane technologies. Certain are nowadays used widely to produce clean water (see next part Membranes for water treatment). The advantage is low saturation of the selective barrier compared to sorbets, as well as decomposition that can take place separately from treated water. The membranes’ function in eliminating ECs from water, usually exposes membranes to a large amount of water that passes through the selective barriers. These classical membranes aim at adequate performance, in terms of rejection, basing the separation mostly on size of the pores. That becomes highly challenging taking in consideration the large diversity of very small ECs. Albert Einstein said: “The world as we have created it is a process of our thinking. It cannot be changed without changing our thinking.” This paper tries to offer unusual, stimulating discussion about different kinds of dense membranes for removal of ECs, confronting the pore size mechanism model with the solution-diffusion model.

## 2. Membranes for Water Treatment (WT)

The membrane processes listed in Table 2 are well known, or have already been mentioned, for WT [56,57,58]. Most of them focus on ion or particle removal.

Certainly, it is easier to transport only one compound, e.g., water, through the membrane than diverse ECs. Table 3 shows that the majority of the most used membrane processes in WT, listed in Table 2, apply water flux and only one EC’s flux. Neither of the two options involves the flux of some other compounds or ions.

The most employed processes, e.g., NF, RO or FO, are supposed to retain contaminants and let the water pass through the selective barrier. Besides Donnan and adsorption, the separation is based mainly on a spherical exclusion mechanism [45,73]. Although defining a membrane pore is controversial, NF uses porous membranes with pore sizes of 1–2 nm, whereas FO and RO use even smaller sizes of less than 1 nm. Such membranes may be very successful for many contaminants [18,88,89,90], but what about ECs? Will these membranes be the most suitable for ECs’ removal, too? It is highly challenging to retain such small ECs with various molecular weights and such different properties. Up to now, the rejection of ECs by NF and RO has reached 40–100% [59,60,61,74,77], depending on the membrane used, the experimental conditions etc. Here, the selectivity vs. effectiveness of membrane separations needs to be elucidated in more detail as follows.

## 3. Selectivity vs. Effectiveness of Membrane Separation

The factors influencing the membrane selectivity, mentioned in reviews dealing with membrane separation and ECs [18,91], are: Charge of the moleculeHydrophobicity of the separated moleculeSize of the separated molecule

Two properties describe the affinity of the molecule to the membrane, which indicate if the molecule is able to diffuse through the membrane or is able to penetrate its pores. The solvent is the permeating species, because of its small molecule size, in the most employed processes such as NF, RO or FO.

A detailed study discussing the factors influencing forward osmosis applied in the removal of ECs was presented by Coday et al. [78]. Their work showed that the experimental conditions may strongly influence the affinity of the target molecule to the membrane. They explained that the mechanism of separation is a combination of electrostatic repulsion, steric hindrance, the solubility and diffusivity of the molecule in the membrane and hydrophilic–hydrophobic forces between the solute and the membrane [78]. According to their work: Rejection of nonionic hydrophilic ECs (e.g., paracetamol, caffeine, methylparaben) is a result of physical sieving.Rejection of hydrophobic nonionic ECs (e.g., carbamazepine, estrone) is most influenced by the initial adsorption of the molecules on the membrane and, as the membrane gets saturated by the solute, the rejection decreases.Positively charged ECs (propranolol, metroprolol) and negatively charged (ibuprofen, naproxen, diclofenac) rejection is connected with the electrostatic interactions between the molecule and the membrane surface, and with sieving.

Considering that the membranes used for osmosis and RO are rather hydrophilic, the message received is not clear: hydrophobic compounds can adsorb on a hydrophilic membrane and saturate it, increasing its hydrophobicity and its permeability for the hydrophobic compounds. This was observed and described in detail by Nghiem et al. [75], who provided a complex study of the dependence of retention of three pharmaceuticals (ibuprofen, sulfamethoxazole and carbamazepine) on the pH of the feed solution, trying to explain the influence of electrostatic interactions between the molecule and the membrane on retention. A relatively hydrophilic TFC-SR2 nanofiltration membrane was used. The zeta potential of the membrane was measured and found to decrease with increasing pH, from +5 mV at pH = 2.5 to −10 mV at pH ≥ 5. It was observed that at pH values where the molecules of the pharmaceuticals were negatively charged (lower than 4), their retention was low (meaning that they were permeating immediately through the membrane due to the strong electrostatic interactions with the positively charged membrane), except for ibuprofen, which was neutral and was adsorbed on the hydrophobic centers.

Acero et al. presented very interesting results illustrating how complicated the situation of the interaction between the separated substance and membrane can be. They used UF membranes with a large molecular weight cut-off (2000 kDa and more) and tested them with a water solution of hydroxybiphenyl. The molecular weight of hydroxybiphenyl is 170 g/mol and it should completely pass through the membrane’s pores if size exclusion was the only mechanism of rejection. However, the rejection of hydroxybiphenyl was nearly 100% because it was completely adsorbed on the membrane. This was proven also by a sorption experiment [59].

Sometimes, authors themselves are confused by their own results. Radjenovic et al. [61] presented precise work using NF and RO membranes, reaching high retention of all the tested substances with RO membranes. The only exceptions were gemifibrozil, and mefenamic acid. The authors’ commented that these compounds are negatively charged and thus should have as high retention as the rest of the negatively charged emerging contaminants (due to electrostatic repulsion), and that there was no plausible explanation. Looking at the figures, it may be seen that the error bars for these two compounds ranged from −20% to 100%, so no explanation is necessary.

Studies on the role of the surface charge of the membrane were presented by Bellona and Drewes as well as by Childress and Elimelech in their much-cited works [76,92]. Childress and Elimelech extended the problem of the feed-solution chemistries, studying also the influence of the presence of surfactants. The electrostatic interactions are also influenced by the presence of background electrolyte. The chemistry of the feed solution always has to be studied when trying to apply membrane separation on ECs removal, because it will probably cause different results in laboratory tests with a model solution compared to tests with real wastewater or wastewater from different sources.

Unfortunately, all the mentioned interactions might be affected by membrane fouling in both the positive and negative senses. Studies connected with membrane fouling and its effect on separation have been published by numerous authors [79,80,81,93,94,95], reporting changes of the membrane surface due to fouling during the RO and nanofiltration processes. The authors found that, besides a decrease of the membrane flux, fouling also causes:Change of the surface charge (zeta potential).Change of the hydrophobicity of the membrane.Adsorption of some trace contaminant in the foulant.Change of the roughness of the membrane surface.

Nghiem et al. [96] worked on four different steroid hormones, tested on double-layered polymeric NF membranes. Average pore size was determined using neutral organic molecules and incorporated steric (size) exclusion out of convection and diffusion effects. Results showed that at the beginning of filtration, adsorption of the hormones to the membrane was the main removal process. Since the membrane pores’ capacity was limited, once the membrane reached adsorption equilibrium, retention rate decreased due to the diffusion of hormones through the membrane. After membrane saturation, the pore size exclusion was less important for the separation compared with that of fresh membranes. A breakthrough curve, used for characterization of adsorption bed, can provide more information about membrane separation mechanisms in general [97]. Even though membrane adsorption reached equilibrium, the breakthrough concentration of the tested hormones remained low due to size excursion mechanisms [96]. Other effects become much more important, like diffusion of solute in the membrane and hydrogen bonding. Furthermore, the adsorbed solute can influence hydrophobicity of the membrane. All of those phenomena can influence the separation and retention qualities of the membrane. NF (membranes with a pore size of 1–2 nm) have shown to be insufficient to treat some ECs [62,63,98]. Thus, further processes are required to fully treat the permeate water of NF. Many NF processes are followed by some AOPs, or other degradation processes, to produce EC-free water [99,100,101,102]. Furthermore, chemical cleaning or regular backwashing of membranes may be a partial solution. However, fouling mitigation, for example antifouling membranes, is very challenging [103]. RO showed promising result in ECs reaching up to 100%. Nevertheless, the process suffers from quick membrane fouling, pretreatment requirement and permeate water being excluded even from essential minerals [90]. This technology is applied to full-scale desalination. Pilot-scale results are more diverse than full scale results, mainly due to the change of hydraulic parameters [61].

Any possibility of worsening the quality of feed water influences the effectiveness of membrane separation. An example is the V&A Waterfront desalination plant, installed as part of Day Zero emergency measures in Cape Town. Notwithstanding the capability of producing 2 ML of potable water per day, its production is still struggling with poorer feed water quality than was originally expected. The selectivity of the membranes turns out to be insufficient because of unexpected pollutants in the feed. These are withdrawn by a huge stream of water, resulting in unsatisfactory quality of the drinking water produced [104].

From the viewpoint of environmental impact assessment, NF and RO show weaknesses in assessing different parameters like global warming potential, nutrient enrichment, fossil fuel usage, ozone depletion or toxicity [105]. NF require much less energy compared to RO [106]. However, NF cannot ensure high rejection for all ECs. RO impact assessment indicates high energy consumption and maintenance requirements [107,108]. Furthermore, the ECs’ lode solution formed after NF or RO processes requires further treatment to reduce environment impact [82].

Considering all the factors influencing membrane selectivity, hydrophobic dense membranes (with target affinities to ECs) could solve the problem of complicated interactions and fouling, with no necessity to push a massive amount of wastewater through the membranes by high-pressure pumps. Despite having many other benefits, there is almost no use of such membrane processes in water depollution from ECs. The authors are unable to say whether this is because the most employed processes, like NF, RO or FO, are automatically tested for ECs elimination, or because obstacles exist that have not yet been overcome; no such publication has been found on the subject. The evaluation of advantages and disadvantages of well-known membrane materials that separate on the basis of particle size, versus dense hydrophobic membranes with target transport of emerging contaminants through a selective barrier, will be presented in next paragraph.

## 4. Dense Membrane Processes with Targeted Transport of ECs

The composition of membranes in WT dealing with ECs is heading towards more and more dense materials (see higher selectivity of more dense RO membranes than NF ones, above). Dense membranes are known for their higher selectivity compared to porous constituents. Their transport process can be described as a solution-diffusion model. Still, even dense RO membranes provide separation based on pore size. Avoiding the permeation of water through the membrane and enhancing the transport of ECs may be achieved by using hydrophobic dense membranes. The authors simplify, in Table 4, the pluses and minuses of membrane separation based on pore size and on target transport of ECs.

**Note (a–c)** Membrane separation based on pore size are the most employed membrane processes in WT such as UF, NF or RO. The processes are well described, membranes are available and the separation is fast. These processes were, however, not primarily designed for removal of ECs. There are several new studies trying to clarify whether UF, NF and RO may be successfully applied in removal of ECs. The rejection of different ECs by NF and RO were from 40 to 100% [59,60,61,74,77] depending on EC. Dense hydrophobic membranes for target transport of ECs (without the pore size mechanism) were not yet used.**Note (d)** The advantage of classical processes, with separation based on pore size, is that only one membrane type is needed for treatment of all contaminants, but insufficient selectivity would lead to additional separation parts. Using dense polymeric membranes for target transport of ECs, several different types would be needed.**Note (e)** Whether the molecule of EC is retained by the RO or NF membrane depends on the characteristics of the molecule, namely its charge, hydrophobicity and size. Which molecules will be retained, and which not, cannot be well predicted because it is not only a matter of sieving (as the method was originally designed). Example: “The molecular weight of the Hydroxybiphenyl is 170 g/mol and it should completely pass through the membrane’s pores if the size exclusion was the only mechanism of rejection. However, the rejection of Hydroxybiphenyl was nearly 100% because it was completely adsorbed on the membrane” [59]. Dense polymeric membranes are predictable for any kind of molecule.**Note (f)** After treatment by RO we get nearly UP water and brine full of different ECs compounds. Such water cannot be released to the environment or drunk [104]. It must be remineralized. No need when using targeted dense membranes because only the contaminants would be removed, and the natural composition of water will remain. In the case of water being contaminated mainly by one or two pollutants, it would be interesting to eliminate only these instead of the entire elimination of the water content (with uncertain results).**Note (g)** Membrane separation based on pore size suffers by fouling that can influence the interactions between the molecule and the membrane and, consequently, also the quality of rejection [81]; see also example in Note (e). The water passes along the dense polymeric membranes and, therefore, no fouling occurs.

The removal of ECs by processes using dense membranes deserves attention of researchers. On one hand, classical technologies, e.g., RO, are commercially ahead and adapting for ECs removal. On the other hand, the membrane processes that may be potentially used for target transport of ECs (applying hydrophobic nonporous membranes), are not yet widely known for such treatment. Still, they are already recognized in other fields such as pervaporation (PV), membrane extraction and petraction, which is a special case of dialysis that uses the nonporous membranes.

### 4.1. Pervaporation

PV is a method for the separation of liquids through a membrane that is in contact with a liquid mixture on the feed side while a vacuum is applied on the permeate side. A phase change of the permeating component occurs as a result of such operation conditions. It enters the membrane as a liquid and leaves the membrane as vapor. The vapor is then usually trapped by refrigeration. Pervaporation processes are widely used for compound separation, especially in aroma and flavors extraction [64,109,110], azeotropic organo-aqua mixture separation, during chemical reactions [111], desalination [65,87], volatile organic compounds (VOCs) removal [66], thermally sensitive substances, dehydration of solvents and compounds recovery. The advantages of this technology are its low energy consumption [112] and separation of trace concentrated solutions [113]. Unlike most known separation methods for filtration, PV and pertraction rely on dense membrane selectivity and involve different separation mechanisms: adsorption in the feed side dissolution, diffusion in the membrane and desorption to the permeate side [70,71]. This nonexpansive separation technology has replaced other processes due to its low energy consumption [67,114], involving sixty-five percent more energy saving than distillation [115] without fouling and intensive maintenance problems. The high selectivity of pervaporation membrane can produce high purity permeate [112]. The permeate is a recovered solute and can be used again thanks to its high purity in the case of a single target compound. PV can also be used for a broad range of aromatic molecules, for example, aroma and flavors extraction or in wastewater purification from trace and persistent compounds [114,116]. The processes do not need high pressure to push huge amounts of water through the membrane as in filtrations processes. A vacuum is applied on the permeate side which creates flux through the membrane in the case of PV, but mainly the processes are based on compound and membrane affinity [117]. PV is now being tested for water desalination [57,68,69,83,84,87].

The rejection ratio, *R*, of the component *i* by the membrane is defined as the fraction of solute kept at the feed side regardless of membrane area or compound concentration. This factor shows a general rejection of the membrane toward the compound at a specific time *t* of the separation. Plotting rejection ratio as a function of time visualizes the breakthrough moment of the membrane [97].
(1)Ri,t=1−Cp,i.tCf,i,t0
where *C*_*p*,*i*,*t*_ and *C*_*f*,*i*,*t*0_ are the concentrations of the compound *i* on the permeate side at the moment *t*, and initial concentration of the feed side, respectively.

The permeation flux *J* of the component *i* through a dense membrane (Equation (2)) is defined as the amount of the permeating substance related to the membrane area (*A*) per unit of time (*t*) [111,118]:(2)$$J_{i}=\frac{n_{i}}{A\;t}$$

The permeation flux is influenced by:(1)Membrane thickness.(2)Driving force.(3)Permeability of the membrane.

The influence of the membrane thickness is easy to understand, the thicker the membrane the lower the permeation flux. Driving force and permeability are more complicated.

Permeability (Equation (3)) is the value that describes the affinity of the membrane material to the permeating component, and vice versa. In the case of the solution diffusion transport mechanism, it is usually defined as a product of the solubility (sorption) coefficient *S* of the component in the membrane and the diffusion coefficient (diffusivity) *D* of the component in the membrane:(3)$$P_{i}=D_{i}\times S_{i}$$

The solubility and diffusivity of the component in the membrane are easily measurable values but they are hard to predict. They are determined by certain interactions and similarities between the membrane and the permeating components, as discussed earlier.

The last thing that influences the PV flux of the permeating component is the driving force. Generally, the driving force is defined as the difference of the chemical potential of the particular component across the membrane. In the case of PV, the chemical potential difference is transformed into the difference between the fugacity of the permeating component in the liquid on the feed side of the membrane (Equation (4)) and the fugacity of its vapor on the permeate side of the membrane in the gaseous phase (Equation (5)).
(4)$$f_{i}^{l}=x_{i}\gamma_{i}p_{i}^{sat}$$
where *x_i_* is the molar fraction of the component in the liquid phase, *γ_i_* is the activity coefficient of the component in the solution and *p_i_^sat^* is the saturated vapor pressure of the component.
(5)$$f_{i}^{g}=y_{i}p(5)$$
$y_{i}$ is the molar fraction of the component in the gaseous phase.

The permeation flux is then expressed as:(6)$$J_{i}=\frac{P_{i}}{l}\left(x_{i}\gamma_{i}p_{i}^{sat}-y^{i}p\right)$$
where $p_{i}$ is the permeability of the membrane for the permeating component and *l* is the membrane thickness [111,118].

As mentioned in the introduction, ECs are a huge group of substances of different kinds. Table 1 shows selected compounds commonly declared to be ECs [13,16,119,120], having differences in their molecular weight (even though compounds of lower molecular weight were chosen), melting point, vapor pressure and water octanol partition coefficient. Their water solubility also varies strongly. From the point of view of PV, compounds with higher vapor pressure and higher water solubility are more suitable because of their higher permeation flux probability. Their vapor pressure and solubility are temperature-dependent, so their values may be influenced by raising the temperature of the membrane or the solution, though such an approach always needs to be considered with respect to the economy of the process.

There have been trials utilizing the pervaporation process for the removal of ECs from water. Higuchi et al. tried to use PV to remove 1,2-dibromo-3-chloropropane (DBCP) from water with a PDMS membrane [85]. The authors performed the pervaporation experiment using an aqueous solution of 10 ppm of DBCP. The membrane area was 15.2 cm^2^ and the membrane thickness was 2 µm. Because of the low vapor pressure of DBCP, the authors decided to heat the membrane to 150 °C to enhance the pervaporation flux. They presented the values of pervaporation flux calculated from the difference between the initial concentration of the DBCP in the feed and the concentration after the pervaporation experiment. They did not say whether anything was found in the cold trap after the PV. The concentration of the DBCP decreased after 10 hours of PV from 10 ppm to 2 ppm, but this could have been caused by the sorption of the DCBP in the membrane only, without passing through, because of the very low concentration of the substance studied. This conclusion is supported by the fact that the same group of authors, one year later, presented work dealing only with the selective sorption of emerging contaminants by a PDMS membrane from water and human milk, where DCBP was one of the studied compounds [121]. Higuchi et al. [86] also tried to remove dioctyl phthalate and butylated hydroxytoluene from mineral water by PV through a PDMS membrane. They used commercially available mineral water in glass and polyethylene terephthalate (PET) bottles. The initial concentrations of ECs in the feed solutions were less than 1 ppm. The separation factor was more or less 100 for all the tested samples. The significant part of this work is the addition of theory and the introduction of a linear correlation between the separation factor of the PDMS membrane, log Kow (“hydrophobicity”) and the vapor pressure of the ECs: the higher the product of the vapor pressure and log Kow, the higher the separation factor.

PV is thus able to separate lighter and more volatile ECs from water. However, it is not suitable for less or nonvolatile ECs, yet, the process is suitable where separation is difficult and other membrane processes are insufficient.

### 4.2. Pertraction

Pertraction is a membrane process using dense or liquid membranes, and the principles are similar to PV. The main difference is that there is liquid on both sides of the membrane: the donor, feed solution on one side and the acceptor solution on the other side. The driving force is also the chemical potential difference, but here it is transformed into the difference of the activity of the particular component across the membrane or, in the simplest case, the difference in the concentration of the permeating component $\left(c_{i}^{d}-c_{i}^{a}\right)$. The permeation flux (Equation (7)) and permeability may be calculated in the same way as in the case of PV.
(7)$$J_{i}=\frac{P_{i}}{l}\left(c_{i}^{d}-c_{i}^{a}\right)$$

At laboratory scale, pertraction is often performed as a batch process that causes a continual decrease of the driving force until the concentrations on both sides of the membrane are equal. A low concentration of ECs will worsen it. This can be solved by using a sweep liquid, which should be recycled in the process (e.g., water that can be redistilled so that residues of the ECs, which have a much higher boiling point, would remain in the solid phase). Perfect stirring is necessary to keep both solutions homogeneous and thus the driving force maximal, and to avoid concentration polarization on both sides of the membrane.

The word pertraction in connection with the separation of pharmaceuticals, ECs, endocrine disruptors etc., is hard to find. But inspiration can be drawn from studies on transdermal permeation of drugs. Such tests are performed with Franz diffusion cells and use thin dense membranes like silicone. Waters et al. [122] tested the transport of five different drugs (benzoic acid, benzotriazole, ibuprofen, ketoprofen and lidocaine), which were chosen because of their different pKa and log Kow, through a silicone membrane at pH values from 4.5 to 8.5. It was found that, due to the hydrophobic character of the membrane, the substances with high pKa (those present mostly in un-ionized form) permeated the membrane much more easily. It was also found that ibuprofen, despite its quite high acidity and being almost fully dissociated at all the tested pH values, showed high permeation fluxes compared to other substances because it is strongly hydrophobic. That was in agreement with previously presented facts. The same research group tested the influence of the presence of surfactants of different types in the feed solution during permeation [123]. They described the possible interactions of the surfactant and the membrane surface, and how the surfactants create a charged layer on the membrane, which affects the electrostatic interactions between the membrane and the permeating component.

A slight terminological diversity occurred during the review process about the interaction of the selected substances of interest [124]. Contemporary ECs are mentioned as trace organic compounds, endocrine disruptors and drugs. An extensive and detailed study was found, performed by Garret et al. in 1968, who, for example, used a key word drug connected with diffusion and polymeric membranes [125,126,127]. The authors presented the diffusion of different drugs through a silastic membrane. This process may also be called pertraction because they used a liquid receiving phase (acceptor solution). The authors also introduced the pH partition hypothesis, which takes into account the fact that only uncharged (un-ionized, not dissociated) compounds interact with the uncharged membrane. This was verified experimentally by the fact that when the pH of the feed solution was equal to the pKa of the studied drug (meaning that one half of the compound was in ionized form), the apparent diffusion constant (permeability) decreased by one half. The previously cited authors also declared that the interaction of the membrane and the permeating component was influenced by the electrostatic interaction and pH of the solution, and a study presented by Garret et al. gave mathematical evidence for this fact. Besides that, the authors screened the permeability of different polymeric membranes for different drugs from different solvents finding, for example, that polyethylene was permeable to acetaminophen in aqueous solution, silastic membrane to acetaminophen from all used solvents and progesterone from ethanol, summarizing it in a simple sentence: “solid polymeric membranes are permeable to uncharged organic molecules which are lipophilic”.

In general, pertraction can be highly selective and thus may not lead to losses of, for example, some important ions from water.

### 4.3. Membrane Extraction

Membrane extraction was developed and tested for analytics. The aim is to increase the concentration of an analyte in a sample by moving it from a worse solvent to a better one. It allows a transfer of the analyte from one solvent to another, avoiding mixing of the two solvents by dividing them with a nonporous membrane. In this case, the molecule is transported to the membrane in the donor phase mainly by convection (shaking, stirring), then the molecule is dissolved in the membrane. It diffuses through the membrane and dissolves in the second solvent. This solvent can be easily selected using solubility or partition coefficient data. Considering the batch mode of the separation, the process can be described as follows.

The membrane flux is given by Fick’s law: (8)$$J_{i}=\frac{D_{i}}{l}\left(c_{water}^{membrane}-c_{organic}^{membrane}\right)$$
where $c_{water}^{membrane}$ is the concentration of the separated compounds on the water phase side of the membrane and $c_{organic}^{membrane}$ on the organic phase side of the membrane.
(9)$$c_{i\;water}^{membrane}=S_{iw}*c_{i\;water}^{bulk}$$
where *S_iw_* is the sorption coefficient of the separated component from the aqueous phase, which is a measurable value.
(10)$$c_{i\;organic}^{membrane}=S_{io}*c_{i\;organic}^{bulk}$$
where *S_io_* is the sorption coefficient of the separated component in the organic phase.

The fact that there are different solvents on both sides of the membrane is reflected also in the mathematical description of the process, because of the different sorption coefficients. It is essential to know both sorption coefficients because this influences strongly the driving force of diffusion and they cannot be hidden in some overall mass transfer coefficient as in PV and pertraction. The factors leading to successful separation are the same as in previous cases: reasonable diffusivity of the component through the membrane and good affinity of the component to the membrane material. Enhancement of the separation here is provided by the better solvent used as an acceptor phase.

Swelling of the membrane in the organic solvent may play an important role in this process. Swelling causes the membrane thickness to increase but also increases the diffusivity of the component in the membrane because the polymeric chains in the membrane material recede from each other. In addition, the organic solvent gets closer to the separated component during swelling.

The solubility of many ECs in other solvents is several orders of magnitude higher than in water.

To illustrate the difference between the solubility of ECs in water and in, for example, ethanol, a simple mathematical exercise may be done. The concentrations of ibuprofen and ethinyl estradiol in two particular studies from the Czech Republic were taken and it was calculated how much of the studied water could be theoretically extracted with 1L of Ethanol. The results are presented in Table 5.

Membrane-assisted extraction is usually mentioned in connection with the analysis of trace compounds in water or environmental samples [128,129,130,131,132]. Concrete examples include the extraction of phenolic compounds from water by silicone rubber [133], UV filters by a low-density polyethylene (LDPE) membrane [134,135], parabens by an LDPE membrane [136], salbutamol and terbutaline by a supported liquid membrane with Aliquot 335 as a carrier [137], and polycyclic musk by an LDPE membrane [138].

A very complex and comprehensive review was presented by Hylton and Mitra [139]. This publication is important because it confirms beyond doubt that using nonporous membranes for the separation of hydrophobic components from water works and is used widely, but in another field than suggested in this present review. Another comprehensive review on membrane extraction in analytical chemistry was published by Jonsson et al. [140]. According to their work, an automated system for nonporous membrane extraction for sample preparation was developed and tested. Although liquid membranes are often used in pertraction and membrane extraction, their employment is certainly not convenient for water purification. There is a risk of possible contamination of purified water by the liquid from the membrane.

Membrane extraction with nonporous polymeric membrane into some convenient solvent could, therefore, be another possible method for removal of ECs from water. Membrane materials such as ordinary LDPE or silicone rubber may be effective enough. The solvent would have to be continually regenerated during the process. On the other hand, only a small amount of solvent (compared to the volume of the feed phase) would be necessary. The advantages and disadvantages are similar to previous alternatives, eliminating ECs from water though membranes. The process would be rather slow, requiring a long contact time and large membrane area. That would be the price for aiming for the highest selectivity and avoiding leakage linked to the immense water flux through a membrane. In addition, the membrane would not be exposed to high pressures. Last, but not least, the target compounds may be removed from the water without losing valuable ions.

## 5. Conclusions

The authors fully agree with Taheran et al. that any method which only creates a more or less concentrated stream, including the membrane separations, cannot be used solely. Still, the popularity of membrane processes in water depollution is great, especially those with elevated water flux through the membranes. This paper compared and discussed a classical pore size mechanism approach of popular nondense (porous) or quasi-dense hydrophilic membranes versus real-dense membranes without pores. It underlines the advantage of, for example, RO being commercially ahead, notwithstanding new adaptation for ECs removal. On the other hand, it confirms beyond doubt that using real-porous membranes for the separation of hydrophobic components from water works, and is used widely, but in another field than suggested here.

Use of real-dense membranes is proposed by the authors as a perspective way to reach the highest selectivity in elimination of ECs elimination. Such membranes are more suitable for flux of ECs through the membrane instead of water Therefore, a hydrophobic separation material would be preferred. Their limitation will surely be connected to the driving force. However, they offer better water/target compounds selectivity than any membrane separating principally on size of the pores (even though they are so small that they are considered negligible). In addition, pressurizing the feed stream would not be necessary. The membranes could be suitable for purposes where high quality takes priority over large quantities. Target transport through the membrane can be suitable for each kind of EC, namely nonionic hydrophilic, nonionic hydrophobic, positively or negatively charged ECs. The real-dense membrane can be well programed for any group of ECs or a specific compound. Knowing the physicochemical properties of the membrane and the separated molecules, appropriate membranes may be employed and allow the emerging contaminants to be concentrated for the next step: safe degradation out of the treated water. The amount of eliminated ECs is not limited as in the case of sorbents. Moreover, concentrating the ECs might help to analyze their composition in water by means of common measurements, which are insufficient when the concentration is too low. As for convenient processes, PV may be used for more volatile compounds, and pertraction and membrane extraction for all ECs.

## Figures and Tables

**Table 1 membranes-10-00305-t001:** Diversity of environmental contaminants (ECs)—physical and chemical properties of selected compounds from the group (e.g., M—molar mass; Log. Kow—octanol/water partition coefficient; MP—melting point; p—pressure; w solub—water solubility).

Category	Name	M (g/mol)	Log Kow	MP (°C)	p sat at 25 °C (Pa)	w solub (mg/L)
Hormones	117-b-estradiol	272.4	4.01	176	8.50 × 10^−7^	3.6
Hormones	Gestodene	270.4	3.13	255		30
Antibiotics	Sulfamethoxazole	253.3	0.89	167	9.23 × 10^−6^	610
Antibiotics	Sulfapyridine	250.3	0.35	192		268
Lipid regulators	Clofibrate	242.7	3.02	<25		insoluble
Lipid regulators	Gemifibrozil	250.3	4.77	62	4.10 × 10^−3^	11
NAIDs	Ibuprofen	206.3	3.97	76	6.30 × 10^−3^	21
NAIDs	Aspirin	180.2	1.19	135	3.60 × 10^−3^	4600
NAIDs	Diclofenac	296.1	4.51	285	8.82 × 10^−5^	2.37
NAIDs	Paracetamol	151.2	0.33	170	8.40 × 10^−3^	14,000
Betablockers	Pindolol	248.3	1.75	170		7880
Betablockers	Propranolol	259.3	3.48	96		61
Anti-depressants	Amitriptiline	277.4	4.81	196	4.80 × 10^−5^	9.71
Anti-depressants	Meprobamate	218.3	0.93	105	4.00 × 10^−1^	4700
Anti-convulsants	Carbamazepine	236.3	13.9	190	2.40 × 10^−5^	17.7
Anti-convulsants	Cabapentine	171.2	−1.10	165	4.00 × 10^−8^	4490
Preservatives	2-phenoxyethanol	138.2	1.16	14	1.30 × 10^0^	24,000
Preservatives	Methylparaben	151.2	1.96	131	3.20 × 10^−2^	2500
Preservatives	Ethyl-4-hydroxy benzoate	166.2	2.47	117	1.20 × 10^−2^	885
Disinfectants	2-phenylphenol	170.2	3.09	60	2.70 × 10^−1^	700
Disinfectants	Chloroprene	88.5	2.2	−130	2.51 × 10^4^	260
Disinfectants	Bromoprene	133		−126	7.35 × 10^3^	
Plasticizers	Diethyl phthalate	222.2	2.47	−4	2.60 × 10^−1^	1080
Plasticizers	Di(2-ethylhexyl) phthalate	390.6	7.6	−55	1.80 × 10^−5^	0.27
Plasticizers	Benzylbutyl phthalate	312.3	4.73	−35	1.10 × 10^−3^	2.69
Plasticizers	Bis(2-ethylhexyl) adipate	370.6	8.1	−70	1.13 × 10^−4^	0.1
Plasticizers	Dibutylphthalate	278.3	4.5	−35	2.70 × 10^−3^	11.2
Plasticizers	1,2-dibromo-3-chloropropane	236.6	2.96	6	1.07 × 10^2^	1.23
Pesticides	Glyphosate	168.1	−3.20	189	1.50 × 10^−5^	1200

**Table 2 membranes-10-00305-t002:** Membrane processes in water treatment (WT).

Process	Nominal Pore Size	Driving Force	Membrane Type	Use
Microfiltration (MF)	0.05–10 µm	Transmembrane pressure diff. 1–3 bar	Porous, as/symmetric	Filtration
Ultrafiltration (UF)	0.001–0.005 µm	Transmembrane pressure diff. 2–5 bar	Microporous asymmetric (PES, TF, CA) *	Filtr. protein and pathogen [59,60]
Nanofiltration (NF)	<2 nm	Transmembrane pressure diff. 5–15 bar	Thin film comp., porous (PA, PS) *	Filtration-large ions [61]
Forward Osmosis (FO)	0.5 nm	Osmotic pressure	Asymm., thin film composite (CTA) *	Desalination [62]
Reverse Osmosis (RO)	0.5 nm	Transmembrane pressure diff. 15–75 bar	Asymm., thin film composite (PA, PBI) *	Desalination [60,61,63]
Electrodialysis (ED)	MW < 200 Da	Electrical potential	Swollen gel, charged, symm.	Desalination
Electrodeionization (EDI)	MW < 200 Da	Electrical potential	Swollen gel, charged, symm.	Desalination
Membrane degasification (MDg)	0.05–0.1 µm	Transmembrane pressure diff., vacuum on perm. side	Porous, symmetric or asymmetric	Degasification
Membrane distillation (MD)	-	Temperature and concentration gradient	Highly porous, symmetric	Desalination
Pervaporation (PV)	-	Transmembrane fugacity difference	Nonporous hydrophilic (CS, SPEEK/PES) * (PDMS/CERAM) * (PEBA/PU) * (PEI/GO) *	Desalination [64,65,66] [67] [68] [69]

* CA—Cellulose acetate; CTA—Cellulose tri-acetate; CS—chitosan; SPEEK—Sulfonated polyether ketone; PES—polyether sulfone; PEBA—polyether block amide; PU—polyurethane; PEI—polyethyleneimine; GO—graphene oxide; PA—Polyamide; PBI—Polybenzimidazole.

**Table 3 membranes-10-00305-t003:** Water and ECs flux through membrane in WT processes from Table 2. Neither of the two options involves the flux of any particles other than water or ECs.

Membrane WT Process	Water Flux	ECs Flux	References
MF	Yes	No	[70,71,72]
UF	Yes	No	[59,60]
NF	Yes	No	[61,73,74,75,76,77]
FO	Yes	No	[62,78,79,80]
RO	Yes	No	[60,61,63,74,81,82]
ED	No	No	Other
EDI	No	No	Other
MDg	No	No	Other
MD	Yes	No	
PV	Yes	Yes	[57,68,69,83,84,85,86,87]

**Table 4 membranes-10-00305-t004:** Pluses and minuses of membrane separation based on pore size and on target transport of ECs through hydrophobic dense membranes.

Perspective	Membrane Separation Based on Pore Size	Separation Based on Target Transport of ECs	Note
Rate of the process	++	--	(a)
Economics	++	--	(b)
Available on the market	+	-	(c)
Number of membrane types needed	+	-	(d)
Prediction	--	++	(e)
Treated water	--	++	(f)
Fouling	-	+	(g)
Pressure	-	+	(h)

++ High advantage; + Advantage; - Disadvantage; -- Important disadvantage.

**Table 5 membranes-10-00305-t005:** Illustration of vast differences between the solubility of ECs (ethinyl estradiol and ibuprofen) in water and ethanol.

Parameters	Units	Ethinyl Estradiol	Ibuprofen
Water solubility	g/L	1.13 × 10^−2^	2.10 × 10^−2^
Ethanol solubility	g/L	50	60
c in particular case	g/L	5.10 × 10^−8^	9.90 × 10^−6^
Reference	-	[77]	[15]
Amount of water from particular case extractable by 1 L of Ethanol	m^3^	9.8 × 10^−5^	6060

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
