# Peer review of "Membrane Removal of Emerging Contaminants from Water: Which Kind of Membranes Should We Use?"

_membranes, 2020, doi:10.3390/membranes10110305_

Round 1
Reviewer 1 Report
This study summarized the advantages/disadvantages of well-known membrane material in water treatment dealing with emerging contaminants. This is a well-organized review, but some remarks should be considered prior to publication on Membranes.
- In Table 2, the authors listed the pore size of various membrane processes, please provided the related references. For instance, the authors said the pore size of ED is MW < 200 Da, this is the size of matter transport across the membrane not the pore size of ion exchange membrane.
- In my opinion, membrane degasification is not feasible for the removal of emerging contaminant.
- In Section 4, the authors highlight three membrane separation processes, i.e., pervaporation, membrane extraction and pertraction. The dominant membrane processes such as NF and RO for the removal of emerging contaminants were not discussed. Do the authors believe that the pervaporation, membrane extraction and pertraction are more competitive than the so-called dense membranes (NF, RO) for the removal of emerging contaminants from water?
- The authors should provide at least one example of each membrane process for the removal of emerging contaminants from water?
- The authors should provide some perspectives about membranes for the removal of emerging contaminants from water and also need to answer the question described in the title.
- The full names of abbreviations should be provided when they are first time appeared in the text, such as the M, Log Kow, MP, etc in table 1.
Reviewer 2 Report
The quality of the manuscript is good and it is clear that the authors put a lot of effort into this work. However, as it is a review paper, the references must be more suitable and clean. Main groups working with membrane technology and new approaches were not cited.
0. Highlights
The first highlight is too long and a bit confusing.
- Introduction
Overall is missing a bit of introduction about membranes. Authors correctly presented ECs and also different treatments, but only in Line 97 membrane technology was presented.
Line 28. There are 3 references for pharmaceutical compounds. However, ECs have many more categories.
Line 31. It is missing a reference about the issue regarding the detection of contaminants at low concentrations.
Check ref:
Diuzheva, H. Dejmková, J. Fischer, V. Andruch, Simultaneous determination of three carbamate pesticides using vortex-assisted liquid–liquid microextraction combined with HPLC-amperometric detection, Microchemical Journal, 150 (2019) 104071.
Lines 31-33: Authors needs to state what are ECs and the categories of ECs, not only pharmaceuticals, EDCs and personal care products. Check References bellow:
J.C. Espíndola, V.J.P. Vilar, Innovative light-driven chemical/catalytic reactors towards contaminants of emerging concern mitigation: A review, Chemical Engineering Journal, 394 (2020) 124865.
Rizzo, S. Malato, D. Antakyali, V.G. Beretsou, M.B. Đolić, W. Gernjak, E. Heath, I. Ivancev-Tumbas, P. Karaolia, A.R. Lado Ribeiro, G. Mascolo, C.S. McArdell, H. Schaar, A.M.T. Silva, D. Fatta-Kassinos, Consolidated vs new advanced treatment methods for the removal of contaminants of emerging concern from urban wastewater, Science of The Total Environment, 655 (2019) 986-1008.
Lines 65 to 75: This is really vague. Authors presented treatment approaches randomly without further comments.
Line 86: The indicated references don’t correspond to the AOPs treatment descried. Photocatalysis and UVC/H2O2 processes, extremely important AOPs were not mentioned and cited.
Check refs:
UVC/H2O2:
Di Cesare, M. De Carluccio, E.M. Eckert, D. Fontaneto, A. Fiorentino, G. Corno, P. Prete, R. Cucciniello, A. Proto, L. Rizzo, Combination of flow cytometry and molecular analysis to monitor the effect of UVC/H2O2 vs UVC/H2O2/Cu-IDS processes on pathogens and antibiotic resistant genes in secondary wastewater effluents, Water Research, 184 (2020) 116194.
J.C. Espíndola, R.O. Cristóvão, S.R.F. Araújo, T. Neuparth, M.M. Santos, R. Montes, J.B. Quintana, R. Rodil, R.A.R. Boaventura, V.J.P. Vilar, An innovative photoreactor, FluHelik, to promote UVC/H2O2 photochemical reactions: Tertiary treatment of an urban wastewater, Science of The Total Environment, 667 (2019) 197-207.
Photocatalysis:
Yao, X. Hu, Y. Liu, X. Wang, X. Hong, X. Chen, S.C. Pillai, D.D. Dionysiou, D. Wang, Simultaneous photocatalytic degradation of ibuprofen and H2 evolution over Au/sheaf-like TiO2 mesocrystals, Chemosphere, 261 (2020) 127759.
A.M. Díez, F.C. Moreira, B.A. Marinho, J.C.A. Espíndola, L.O. Paulista, M.A. Sanromán, M. Pazos, R.A.R. Boaventura, V.J.P. Vilar, A step forward in heterogeneous photocatalysis: Process intensification by using a static mixer as catalyst support, Chemical Engineering Journal, 343 (2018) 597-606.
Others:
Stathoulopoulos, A.; Mantzavinos, D.; Frontistis, Z. Coupling Persulfate-Based AOPs: A Novel Approach for Piroxicam Degradation in Aqueous Matrices. Water 2020, 12, 1530.
- Membranes for water treatment
A column with membrane material must be added in Table 2.
- Selectivity vs. effectiveness of membrane separation
Lines 202-211: The “breakthrough curve” phenomenon needs to be stated. There are many references about it.
Lines 213-215: “Thus, further processes will be required to fully treat the water by NF. Chemical cleaning or regular backwashing of membranes may be a partial solution; however, fouling mitigation is very challenging”
This phrase is misleading. Chemical cleaning or regular backwashing is a solution to fouling. However, authors were talking about the inefficiency of NF to treat water, as other processes were needed and NF is suitable for ECs rejection not degradation.
Author should cite works that couple membrane technology with photochemical processes. The groups of Prof. Sylwia Mozia and Prof. Falaras Polycarpos have been working with it. Deeper research must be done.
- Selectivity vs. effectiveness of membrane separation
The heading 4 has the same name of heading 3. Please correct it.
Lines 320-335: Would be interesting also add “rejection” calculation.
List of abbreviations
AOP: advanced oxidation process
